# Evaluating Mortality Response Associated with Two Different Nordic Heat Warning Systems in Riga, Latvia

**DOI:** 10.3390/ijerph17217719

**Published:** 2020-10-22

**Authors:** Kerstin Pfeifer, Daniel Oudin Åström, Žanna Martinsone, Darja Kaļužnaja, Anna Oudin

**Affiliations:** 1Division of Occupational and Environmental Medicine, Department of Laboratory Medicine, Lund University, 22381 Lund, Sweden; daniel.astrom@envmed.umu.se (D.O.Å.); anna.oudin@med.lu.se (A.O.); 2Institute of Occupational Safety and Environmental Health, Riga Stradins University, LV-1007 Riga, Latvia; Zanna.Martinsone@rsu.lv; 3Department of Occupational and Environmental Medicine, Riga Stradins University, LV-1007 Riga, Latvia; Darja.Kaluznaja@rsu.lv; 4Division of Sustainable Health, Umeå University, 901 87 Umeå, Sweden

**Keywords:** heat waves, mortality, heat warning systems, Latvia

## Abstract

*Background and objectives*: Progressing climate change is accompanied by a worldwide increase in the intensity, frequency, and duration of heat wave events. Research has shown that heat waves are an emerging public health problem, as they have a significant impact on mortality. As studies exploring this relationship are scarce for Latvia, this study aims to investigate the short-term associations between heat waves and all-cause mortality as well as cause-specific mortality, during the summer months (May-September) in Riga. *Materials and Methods*: An ecological time series study using daily reported mortality and temperature data from Riga between 2009 and 2015 was employed. Heat waves were defined based on the categories of the Latvian and Swedish heat warning system. Using a Quasi-Poisson regression, the relationships between heat waves and all-cause as well as cause-specific mortality were investigated. *Results*: Heat waves in Riga were associated with a 10% to 20% increase in the risk of all-cause mortality, depending on the applied heat wave definition, compared to days with normal temperature. In addition, heat-related mortality was found to increase significantly in the ≥65 age group between 12% and 22% during heat waves. In terms of cause-specific mortality, a significant increase of approximately 15% to 26% was observed for cardiovascular mortality. No significant associations were found between heat waves and respiratory or external causes of mortality. *Conclusion*: These results indicate that there are short-term associations between heat waves and all-cause as well as cardiovascular mortality in Riga and that heat waves therefore represent a public health problem in this Baltic city.

## 1. Introduction 

Due to climate change, the global average temperature is rising, with the greatest changes being observed at middle and high latitudes in the Northern Hemisphere [1]. This change is accompanied by periods of extreme heat, often referred to as heat waves, which are a natural hazard and an emerging public health problem [2]. Worldwide, heat waves have been observed to be increasing in their frequency, intensity, and duration [2,3,4]. These increases give rise to public health problems, as heat waves can take a heavy toll on human and natural systems, affecting livelihoods, infrastructure, and health [5].

There is a substantial body of evidence suggesting increased mortality and morbidity during heat waves for European countries [6,7,8,9,10,11]. In the greater Baltic Sea Region, increasing mortality due to elevated temperatures has been reported in multiple countries such as Lithuania [12], Finland [13], Estonia [14], Sweden [15,16], and Poland [17]. These studies indicate that heat and heat waves lead to an increase in mortality in this climatic region, despite these countries having low average temperatures, suggesting that relatively high temperatures as compared to more normal temperatures may be of importance. Previous research has suggested region-specific impact of heat, with heat-related mortality occurring at higher temperatures in warmer regions [18]. Populations residing in cities with milder summers were more susceptible to heat than inhabitants in cities with higher summer temperatures [15,19].

Heat warning systems (HWS) have been introduced in numerous countries worldwide, as HWS are generally seen as an important tool to minimize the negative health impacts of heatwaves, including heat-related mortality [20,21]. For example, after the implementation of a new HWS in Frankfurt (Germany), lower excess mortality was measured in the following years [22], and in France it was estimated that about 4400 premature deaths were prevented during the 2006 heat wave due to the introduction of a HWS in 2004 [23]. Heat warning systems serve to prevent negative health impacts by providing alerts to the health sector and the general population when a critical threshold temperature is reached [20].

In Latvia a HWS exists [24]; however, studies on the association between heat and mortality are rare. A study by Merte [25] estimated, using country-level data, that heat waves cause about 84 (±19) excess deaths per year. However, research has shown that the relationship between temperature and health varies from region to region [5,9] and that respiratory and cardiovascular diseases have a great impact on heat-related mortality [6,26]. It is therefore important to investigate the local situation in Latvia, such as in the capital Riga, and to also analyze cause-specific mortality.

The aim of the present study is to investigate all-cause as well as cause-specific mortality during heat waves in Riga, Latvia, using different regional thresholds for defining a heat wave.

## 2. Materials and Methods

Situated in northeastern Europe, Latvia and its capital Riga belong to the northern part of the temperate climate zone and are located in a transition zone between maritime and continental climates (Figure 1).

Daily mean and maximum temperature data for Riga were supplied by the Latvian Environment, Geology and Meteorology Centre for the five warmest months of the year (from May to September) with no missing observations for the study period (2009–2015). Daily all-cause mortality data for the same period were provided by the Latvian Register of Causes of Death.

As no universal definition for heat waves exists, two different heat warning systems from the Baltic region, more specifically from Latvia and neighboring Sweden, were used to define the heat wave categories.

The Latvian heat warning system consists of two categories [24]. The first level warns when daily maximum temperatures are between 27 °C and 32 °C for at least two consecutive days or longer. The next warning level is issued if daily maximum temperatures exceed ≥33 °C. Based on these definitions, the two heat wave categories will be referred to as LVA.HW_1_ and LVA.HW_2_ (see Table 1).

Based on the Swedish heat warning system, three heat wave categories were defined [28]. In Sweden, a first warning message for high temperatures is sent out when daily maximum temperatures reach at least 26 °C and up to 30 °C for three consecutive days. In this study, 27 °C was used as a lower threshold for the first category, as explained in the study by Åström et al. [15], and no upper threshold was used. Keeping these category definitions in line with the existing Swedish study allows for later comparison. In order for a first-class warning for very high temperatures to be issued, daily maximum temperatures need to be at least 30 °C for three consecutive days. For a second-class warning for extremely high temperatures to be issued, the daily maximum temperatures need to be 30 °C or higher for five consecutive days and/or daily maximum temperatures at least 33 °C for three consecutive days. The three heat wave categories based on the Swedish definitions will be referred to as SWE.HW_1_, SWE.HW_2_, and SWE.HW_3_ (see Table 1).

All-cause mortality was used in the main analysis as external-cause mortality due to heat being present in the region, and excluding it may underestimate the total effect of heat waves on mortality [29]. We also analyzed daily mortality due to respiratory (ICD 10 J00–J99), cardiovascular (ICD 10 I00–I99) and external-cause (ICD 10 V00–Y99) mortality, as well as stratifying the analyses on age groups (above and below the age of 65).

As overdispersion was present in our daily counts of mortality, the hypothesized short-term association between heat waves and mortality was analyzed using an overdispersed Poisson regression model. Time trends were controlled for using a natural cubic spline which was allowed four degrees of freedom per year of study and additionally controlled for day of week patterns in mortality with a categorical variable. To statistically test if age or heat wave definition modified the effects of heat waves, we calculated the ratio between the relative risk (RR) of the group of interest and the RR from the reference group [30]. We chose not to include air pollution in the models as air pollution is suggested to be a mediator in health-effect studies of temperature [31]. In addition, humidity seems not to have a large impact on associations between heat waves and mortality in epidemiological studies. A recent study conducted in 24 countries and 445 cities reported absence of a positive association of humidity with mortality in summer [32].

Model checks were performed by the visual inspection of normally distributed residuals, see Appendix A for an example of residual checks. All analyses were performed with R version 3.6.1 (R Core Team, Vienna, Austria).

All results are presented as RRs for the LVA.HW_1_, SWE.HW_1_, and SWE.HW_2_, along with their corresponding 95% Confidence Intervals (CI). The mortality associated with LVA.HW_2_ and SWE.HW_3_ were not analyzed as there were no heat wave events for these heat wave categories within the study period.

## 3. Results

### 3.1. Descriptive Statistics

Table 2 presents the daily maximum temperature and the number of heat wave days during the warmest months in Riga.

The warmest month during the study period was July with a mean temperature of 19.8 °C and a maximum temperature of 32.5 °C, followed by August (mean/maximum temperatures 18.1/32.9 °C). These two months also had the most heat wave days, as July had a total of 44 heat wave days and August had a total of 13 heat wave days, when combining the three investigated heat wave categories. In general, using the Latvian heat wave definition, 8 additional heat wave days were measured compared to the first Swedish heat wave category. SWE.HW_2_ only occurred in Riga from 10 to 13 July 2010 and from 3 to 5 August 2014 when the city experienced three consecutive days with temperatures exceeding 30 °C.

Descriptive statistics of the daily mortality counts are presented in Table 3.

In total, out of an average population size of 656,877, there were 24,273 deaths in Riga during the study period, of which about 73% occurred in the age group 65 years and older. Of the three specific causes of mortality investigated, cardiovascular causes were most prevalent, making up about 53% of mortalities.

### 3.2. Relative Risks Associated with Heat Waves

Table 4 presents the relative risks of mortality during the different definitions of heat waves for the six mortality and age groups.

Strong evidence of increased mortality was found for all-cause mortality during HW_1_ events for both the Latvian and Swedish HWS. LVA.HW_1_ was associated with an RR of 1.1 (95% CI: 1.02–1.18) and SWE.HW_1_ with an RR of 1.2 (95% CI: 1.11–1.31) for all-cause mortality. In addition, heat-related mortality in the ≥65 age group also increased significantly for both LVA.HW_1_ and SWE.HW_1_ compared to non-heat wave days, with RRs of 1.12 (95% CI: 1.02–1.21) and 1.22 (95% CI: 1.11–1.33), respectively. The RRs for the above 65 age group were higher than for those under 65 but not to a statistically significant extent. In terms of cause-specific mortality, cardiovascular mortality was observed to significantly increase during heat waves for both LVA.HW_1_ (RR 1.15, 95% CI: 1.05–1.27) and SWE.HW_1_ (RR 1.26, 95% CI: 1.14–1.41). No significant association was found between heat waves and respiratory or external causes of mortality.

In addition, mortality during heat waves according to SWE.HW_2_ increased, except for respiratory causes, but not to a significant level. When comparing the two Swedish heat wave categories, the heat wave mortality burden was, for most of the mortality groups, higher during the first category compared to the second category; however, this result was not statistically significant.

## 4. Discussion

In this time-series analysis of the summer months May to September from 2009 to 2015, we found strong evidence between heat waves and increased all-cause mortality. In Riga, two consecutive days of temperatures between 27 °C and 32 °C, which occurred 37 times, increased all-cause mortality by approximately 10% compared to non-heat wave days.

Adding one consecutive day to the heat wave definition with temperatures above 27 °C increased the risk of all-cause mortality from 10% to 20%. In general, the relative risks of mortality were higher when applying the first Swedish heat wave category compared to the first Latvian one. However, an increase in heat intensity from SWE.HW_1_ to SWE.HW_2_ did not result in an increase in relative risk. Using the same Swedish heat wave definitions as in this present study, Åström et al. reported an increase of 8% and 15% for all-cause mortality for the first and second heat wave category, respectively [15]. In Sweden, an increase in the heat wave intensity increased the mortality risk. This effect was not present in this study, likely due to the low number of SWE.HW_2_ events identified in Latvia within the study period.

When it comes to cause-specific mortality, this study confirms that heat waves are associated with deaths due to cardiovascular causes, as these increased significantly in Riga by 15–26% during heat waves. Mortality rates due to cardiovascular diseases are generally higher in Latvia compared to other countries in the EU [33]. The finding that cardiovascular mortality is increased during heat waves is in line with other recent studies in Sweden and Poland, which also report increases in mortality due to cardiovascular diseases [15,17,34]. Hot temperatures have been found to increase cardiovascular strain as a result of increased body temperature or thermoregulatory processes in humans. This additional strain on the body may lead to excessive cardiovascular strain and deaths [35,36]. Cardiovascular response to heat may differ between healthy and vulnerable populations as well as between gender [37,38].

In addition to cardiovascular causes of mortality, respiratory and external causes of mortality were also investigated in the present study, but with inconclusive and non-significant results. On the contrary, a review on respiratory effects of heat waves has found an overall significant positive association between heat waves and respiratory mortality [39]. The underlying physical mechanisms behind this association are not yet completely understood, but airway hyper-responsiveness and impaired thermoregulation leading to hyperventilation and exacerbations could be a possible trigger for respiratory problems [40,41]. One possible explanation for the non-significant results on respiratory mortality in this short-term analysis is the suggested evidence, provided in studies on the effects of heat waves and air pollution on mortality, that cardiovascular diseases can initiate an acute body response during heat waves, whereas respiratory diseases are slower in their development and therefore more lagged [42]. Regarding external-cause mortality, such as traffic accidents, fire, assault, and drowning, Orru and Åström [29] reported that shortly after exposure to elevated temperatures, mortality due to external causes increased significantly in Estonia. Thus, even if the present study can give a clear picture only of the relationship between cardiovascular mortality and heat waves in Riga, other studies indicate that a positive association between heat waves and respiratory as well as external causes of mortality may be present.

The results of this study support evidence from previous observations that being elderly (≥65 years) is associated with a higher mortality risk during heat waves [17,34,43], although the effect did not differ significantly between the two investigated age groups. A higher susceptibility to heat waves among older individuals is based on both social and medical underlying factors [44,45]. Thermoregulation might be impaired due to medication or the deterioration of physiological mechanisms in the body which occur with aging [44]. In addition, a reduced ability to protect oneself from heat stress and living alone are risk factors among the elderly [44,45,46].

In the broader European setting, similar impacts of heat waves on mortality can be observed. The EuroHEAT project compared data from nine European cities and reported a significant increase in total daily mortality related to heat waves and suggested that daily mortality increased with age [6]. Contrary to the data in Riga, in most of the other nine European cities, the increase of respiratory mortality was greater than cardiovascular mortality [6]. Nonetheless, there was great heterogeneity between the cities, which indicates that the effect of heat waves differs from region to region.

As Riga is a large city, the results of the study are likely to differ from regional areas due to the urban heat island effect. Thus, during heat waves, temperatures are likely to be higher in Riga compared to regional areas [5]. Moreover, because there is no universal heat wave definition, heat waves can be defined in very different ways across different regions, and thresholds for heat waves vary depending on the local response to extreme heat. All the different definitions make it difficult to compare results from different studies, and therefore comparisons, including the above, should always be considered with caution. To generalize the results to other parts in Latvia would require more comprehensive data and analyses across multiple cities in different climatic zones in Latvia, in addition to estimations of the heat wave impact on rural populations.

### Strengths and Limitations

A strength of this study is that in addition to all-cause mortality, cardiovascular, respiratory, and external causes of death were investigated. The validity of the Latvian death register can be considered high, as medical certificates of death are compiled by medical professionals in Latvia [47], and consequently misclassification of deaths should be limited.

This study also has several limitations. The use of aggregated data neglects potential differences in mortality occurrence at the city district level. In addition, this study investigated only the short-term effects of heat waves and analyzed mortality on heat wave days compared to non-heat wave days. While most deaths occur within the first days of heat waves, some deaths due to heat can also be observed several days later [3], and these were not captured in the present analysis.

Other limitations are related to the available data. The length of the time series is, compared to some other studies, short. However, Armstrong et al. provided a good overview of sample size issues in time series regressions, and we believe that the current study is sufficiently powered to be able to draw valid conclusions regarding the impact of heat waves on all-cause mortality in Riga, Latvia [48]. The data on respiratory and externally induced mortality do not provide sufficient evidence to come to an unambiguous conclusion on their association with heat wave-related mortality in the setting of Riga. Furthermore, as no data on sex were available, it was not possible to run sex-specific analyses to investigate whether sex modified the association between heat waves and mortality. Earlier studies suggest that sex can be an effect modifier and that in some locations one sex is more susceptible than the other one [6,11,49,50]. In addition, data from previous studies have shown that HWSs, often implemented as part of broader heat-health action plans, can have positive effects on the relationship between heat waves and mortality and have the potential to help prevent premature deaths [20,21,22,23,51]. However, this study does not provide enough information to evaluate the Latvian heat warning system and draw conclusions on its effect on heat-related mortality in Riga.

## 5. Conclusions

This study examined the impact of heat waves on mortality in Riga during the months May to September from 2009 to 2015. Even though existing heat warning systems in the region warn at different levels, mortality increased to a similar extent. Nevertheless, it was found that the mortality risk during the first category of the Swedish HWS was slightly higher for most mortality groups than during the first category of the Latvian HWS. The results indicate that for both HWS, all-cause mortality as well as cardiovascular mortality increase significantly when temperatures rise above 27 °C for at least two or three days. A significantly increased mortality due to periods of extreme heat was also found in the ≥65 age group when comparing heat wave days to non-heat wave days. These results are similar to those of other countries within the Baltic Sea Region and Europe and show that heat waves are a serious public health concern also in Riga. Further research should be carried out to assess whether the HWS thresholds are well chosen and whether the results from Riga are comparable to other parts of the country. In addition, further action should be taken to ensure that the Latvian HWS is incorporated into a broader heat-health action plan to prevent negative health impacts of heat waves in the future.

## Figures and Tables

**Figure 1 ijerph-17-07719-f001:**
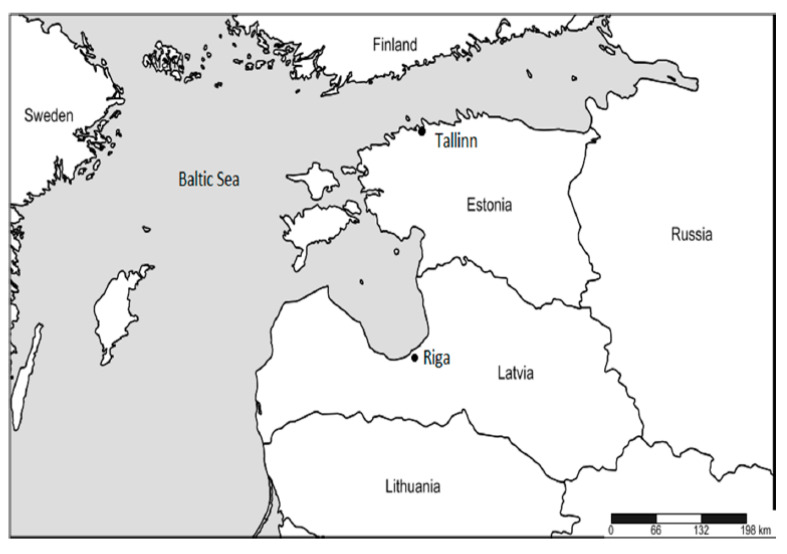
The study area and neighboring countries [27].

**Table 1 ijerph-17-07719-t001:** Overview of heat wave variable definitions.

Heat Wave Category	Definition
LVA.HW_1_	Two consecutive days with daily maximum temperature between 27 °C and 32 °C.
LVA.HW_2_	Daily maximum temperature ≥33 °C.
SWE.HW_1_	Three consecutive days with daily maximum temperatures >27 °C.
SWE.HW_2_	Three consecutive days with daily maximum temperatures >30 °C.
SWE.HW_3_	Five consecutive days with daily maximum temperatures >30 °C and/or three consecutive days with daily maximum temperatures >33 °C.

**Table 2 ijerph-17-07719-t002:** Daily mean, maximum temperatures and number of heat wave days for the summer months May to September from 2009 to 2015 in Riga.

Period	Daily Maximum Temperature					*n* Heat Wave Days ^b^		
	Mean	SD ^a^	Median	Min	Max	LVA.HW_1_	SWE.HW_1_	SWE.HW_2_
May–September	19.1	5.0	18.8	5.5	32.9	37	29	3
May	16.2	5.1	15.8	5.9	29.3	3	2	0
June	19.0	4.1	18.8	10.8	30.3	5	2	0
July	22.9	4.5	22.9	14.6	32.5	24	18	2
August	20.9	3.9	20.5	13.5	32.9	5	7	1
September	16.3	3.4	16.5	5.5	24.0	0	0	0

^a^ SD: Standard deviation ^b^ Number of heat waves according to the levels of heat waves of the two different heat wave warning systems. LVA.HW_1_: Two consecutive days with daily maximum temperature between 27 °C and 32 °C; SWE.HW_1_: Three consecutive days with daily maximum temperatures >27 °C; SWE.HW_2_: Three consecutive days with daily maximum temperatures >30 °C.

**Table 3 ijerph-17-07719-t003:** Average mortality data for the summer months for the period 2009–2015 in Riga.

Cause of Mortality or Age Group	Number of Deaths in Age Group	Annual Summer Mortality Rate (per 1000 Inhabitants)	Daily Number of Deaths				
			Mean	SD	Median	Min	Max
Total	24,273	5.3	22.7	5.2	23	9	41
≤64	6486	1.4	6.1	2.5	6	0	18
≥65	17,787	3.9	16.6	4.4	16	5	35
Cardiovascular	12,746	2.8	11.9	3.6	12	3	28
Respiratory	616	0.1	0.6	0.8	0	0	5
External	1546	0.3	1.4	1.3	1	0	6

**Table 4 ijerph-17-07719-t004:** Relative Risks (RRs) of mortality with 95% confidence intervals associated with heat waves during the summer months 2009 to 2015 in Riga.

Mortality	LVA.HW_1_	SWE.HW_1_	SWE.HW_2_
All-cause	1.10 (1.02–1.18)	1.20 (1.11–1.31)	1.16 (0.93–1.46)
≤64	1.05 (0.91–1.21)	1.17 (1.01–1.36)	1.43 (0.97–2.11)
≥65	1.12 (1.02–1.21)	1.22 (1.11–1.33)	1.07 (0.81–1.41)
Cardiovascular	1.15 (1.05–1.27)	1.26 (1.14–1.41)	1.23 (0.90–1.66)
Respiratory	0.82 (0.50–1.34)	0.83 (0.48–1.42)	0.92 (0.21–3.93)
External	0.97 (0.73–1.30)	0.93 (0.67–1.29)	1.13 (0.51–2.50)

LVA.HW_1_: Two consecutive days with daily maximum temperature between 27 °C and 32 °C; SWE.HW_1_: Three consecutive days with daily maximum temperatures >27 °C; SWE.HW_2_: Three consecutive days with daily maximum temperatures >30 °C.

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
