# Peer review of "Evaluating Mortality Response Associated with Two Different Nordic Heat Warning Systems in Riga, Latvia"

_ijerph, 2020, doi:10.3390/ijerph17217719_

Round 1

Reviewer 1 Report

Thanks dear author for the study. Welcome to your response and efforts given in this version. 

Still I am on my points as it is related with a single regional coverage. Moreover, five months coverage of temperature explicitly and implicitly diminishes the big scope of observations particularly. 

In this stage contribution is the major issues as many studies covered a variety of findings considering more sample, regional coverage and time period.

Author Response

Thanks dear author for the study. Welcome to your response and efforts given in this version.

Still I am on my points as it is related with a single regional coverage. Moreover, five months coverage of temperature explicitly and implicitly diminishes the big scope of observations particularly. In this stage contribution is the major issues as many studies covered a variety of findings considering more sample, regional coverage and time period.

Author response: Regarding the limited time period investigated we agree with the reviewer that having additional years of data would strengthen the study.

Investigating the impact of heat in a single city is common practice within the field, and no such investigations are currently available for Riga, Latvia. We are aware that recently a lot of publication have been generated using multi-county, multi-city datasets, however we believe that it is worth presenting results from a city where no such investigations have been performed previously. The implications of not allowing single city estimates to be generated would result in publication bias. Generalisability of the results to other populations is then something different. Again, we do not claim that the present study is representative of Latvia, or the Baltics, as we clearly state in the limitation section.

As for the choice of using only the summer months, this approach is suitable when investigating the impact of heat on mortality. We are comparing heat wave days to normal summer days, which presumably are hotter than days in the winter, instead of heat wave days to any other day of the year, be it in August or January.

Reviewer 2 Report

The author evaluates the mortality due to heat waves, using two warning systems; Latvia's and Sweden's. The outcomes are well presented, but the methodology description lacks in terms of description and assumptions made. The justification of quasi-poisson regression use and a hypothesis testing to validate the results, are missing. 

In general, the correlation and causality between heat and mortality is well established. The persistent hot days creates the conditions to observe that in real life. The latter is also a well established fact. By using two different filters (latvia&sweden definition of heat waves), the author creates two samples of data on which he performs a regression analysis. The results are basically the same, as it should, because small different in heat wave definition could not give different result but a variance to the overall conclusion, that heat waves increase mortality. As such, I fail to see the novelty of the current paper.

Author Response

The author evaluates the mortality due to heat waves, using two warning systems; Latvia's and Sweden's. The outcomes are well presented, but the methodology description lacks in terms of description and assumptions made. The justification of quasi-poisson regression use and a hypothesis testing to validate the results, are missing.

Author response: Thank you for pointing this out to us. We have added a description regarding the use of quasi-poisson regression in the revised version of the manuscript.

The sentence on page 2, line 108 now reads: “As overdispersion was present in our daily counts of mortality, the hypothesized short-term association between heat waves and mortality was analyzed using an overdispersed Poisson regression model”.

Regarding hypothesis testing, we added that when comparing the results for effect modifiers or differences between the heat warning systems we used the relative effect modification index.

The following sentence was added on page 2, line 112 and 113: “To statistically test if age or heat wave definition modified the effects of heat waves we calculated the ratio between the RR of the group of interest and the RR from the reference group”.

In general, the correlation and causality between heat and mortality is well established. The persistent hot days creates the conditions to observe that in real life. The latter is also a well established fact. By using two different filters (latvia&sweden definition of heat waves), the author creates two samples of data on which he performs a regression analysis. The results are basically the same, as it should, because small different in heat wave definition could not give different result but a variance to the overall conclusion, that heat waves increase mortality. As such, I fail to see the novelty of the current paper.

Author response: We agree with the reviewer that  the impact of heat on mortality is well established. We do not claim that the present study presents novel evidence that heat impacts mortality, merely adding to the vast scientific literature published. The main contribution of the present study is that it adds further evidence to that relationship, in a city where previously no published studies are available. We therefore believe that our results should be communicated. We would like to draw an analogy to the thousands of air pollution studies that have been published in recent years, to better understand the relationship between air pollution and various health outcomes, even though it has been established for long that air pollution is detrimental for health. As many heat-related deaths are avoidable, our results may help local decision makers when implementing heat health action plans in the region. We believe that locally derived estimates in a real life setting serve better to inform local heat health action plans than estimates from other regions. Furthermore, even if the different heat wave definitions provide similar results, it is, from a local public health perspective, of interest that the point estimates for the Latvian heat warning system are lower.

Round 2

Reviewer 2 Report

Overall, my points still stand.

The author evaluates the mortality due to heat waves, using two warning systems; Latvia's and Sweden's. The outcomes are well presented, but the methodology description still lacks in terms of description and assumptions made. The manuscript lacks the a hypothesis testing to validate the results, are missing. The RR (I assume Relative Risk as it is not stated in the manuscript) is a probabilistic measure not a hypothesis test. The hypothesis testing aims to assess whether the results (RR among them) contain enough information to cast doubt on conventional wisdom.

Moreover, I still fail to see any novelty to the manuscript that would justify the publication to a scientific journal.

This manuscript is a resubmission of an earlier submission. The following is a list of the peer review reports and author responses from that submission.

Round 1

Reviewer 1 Report

I have gone through the study; it is very impressive and timely because of the issue of global warming. In my opinion, the study is limited by time and sample selection. Still, now there are some influential studies conducted based on temperature and many other issues including income level, public health, morality.

The study covered a very limited time that is not enough to judge the historical impact of temperature. Another issue is sample coverage. It covered only a region of a country that is also not suitable for judging the impact of temperature on mortality. The contribution of the study is very limited and narrow. Therefore, I suggest to come up with longitudinal data. 

Diffenbaugh, N. S., & Burke, M. (2019). Global warming has increased global economic inequality. 125, Proceedings of the National Academy of Sciences, 116(20), 201816020.

Kulp, S. A., & Strauss, B. H. (2019). New elevation data triple estimates of global vulnerability to sea-level rise and coastal flooding. Nature Communications, 10(1). doi:10.1038/s41467-019-12808-z

Reviewer 2 Report

Very interesting article studying the association between short-term heat-waves and mortality. Data from 2009 - 2015 was monitored for summer months (May-Sept). Up to 20% increased risk of mortality was discovered, with >=65 age group being particularly vulnerable. Also address an important knowledge gap as lack of studies into heat waves and mortality in Latvia. I have very minor comments, which are listed below:

page 1 - second paragraph: Are greater Baltic regions more susceptible than other latitudinal locations? This paragraph seems to suggest a greater susceptibility of citizens due overall lower average temperatures? Can the authors compare and contrast statistics with other geographic locations to substantiate this interesting hypothesis?

page 2 - first paragraph: Are there universal HW criteria?

page 2 - third paragraph: what are the characteristic differences between all-cause and cause-specific mortality.

page 5 - last paragraph: can the authors cite physiological studies (with model vertebrate organisms - mice, rats, fish) that experimentally demonstrate effects of increased temp on cardiovascular strain. Citing 2-3 studies will give these associations more credibility.

Can heat waves be exacerbating other the effects of other causal factors for the increased risk of mortality - such as particulate pollution etc. Some discussion towards this end is provided for respiratory stress in page 6, first paragraph.

page 6 - first paragraph: what are the characteristics or causes of 'mortality due to external causes'?

Reviewer 3 Report

This paper presents the relationships between the heatwaves and mortality in the Riga, Latvia. I believe this paper is well-written and well-structured.

  1. The introduction is clear and well-written. It basically introduces the global warming and heatwaves, the impact of heatwaves (heat-induced mortality) in other countries and then indicate the research gap of this paper. 
  2. The research methods have been well presented with an explicit definition of heatwaves in Sweden and Latvia given in the heat warning system.
  3. The results are clearly presented and the statistic analysis is robust to use. At the same time, authors have indicate the strength and weakness of this paper. Overall, authors have meet the research aim and objectives this paper set.

I have two minor questions about this paper.

  1. Authors may separate the abstract of this paper into four paragraphs according to the subtitles.
  2. Authors may further explain "as July had a total of 44 heat wave days and August had a total of 13 heat wave days" (Page 4), since readers cannot read such results based on your tables and figures.